# Synthetic Glabridin Derivatives Inhibit LPS-Induced Inflammation via MAPKs and NF-κB Pathways in RAW264.7 Macrophages

**DOI:** 10.3390/molecules28052135

**Published:** 2023-02-24

**Authors:** Jaejin Shin, Leo Sungwong Choi, Hyun Ju Jeon, Hyeong Min Lee, Sang Hyo Kim, Kwan-Woo Kim, Wonmin Ko, Hyuncheol Oh, Hyung Soon Park

**Affiliations:** 1Glaceum Inc., Suwon 16675, Republic of Korea; 2Department of Applied Chemistry, Institute of Natural Science, Global Center for Pharmaceutical Ingredient Materials, Kyung Hee University, Yongin 17104, Republic of Korea; 3Department of Herbal Crop Research, National Institute of Horticultural and Herbal Science, RDA, Eumseong 27709, Republic of Korea; 4Department of Marine Bio-Food Sciences, Chonnam National University, Yeosu 59626, Republic of Korea; 5College of Pharmacy, Wonkwang University, Iksan 54538, Republic of Korea

**Keywords:** glabridin derivatives, HSG4112 (vutiglabridin), HGR4113, LPS-stimulated macrophage, anti-inflammation, NF-κB

## Abstract

Glabridin is a polyphenolic compound with reported anti-inflammatory and anti-oxidative effects. In the previous study, we synthesized glabridin derivatives—HSG4112, (S)-HSG4112, and HGR4113—based on the structure–activity relationship study of glabridin to improve its biological efficacy and chemical stability. In the present study, we investigated the anti-inflammatory effects of the glabridin derivatives in lipopolysaccharide (LPS)-stimulated RAW264.7 macrophages. We found that the synthetic glabridin derivatives significantly and dose-dependently suppressed the production of nitric oxide (NO) and prostaglandin E2 (PGE2), and decreased the level of inducible nitric oxygen synthase (iNOS) and cyclooxygenase-2 (COX-2) and the expression of pro-inflammatory cytokines interleukin-1β (IL-1β), IL-6, and tumor necrosis factor alpha (TNF-α). The synthetic glabridin derivatives inhibited the nuclear translocation of the NF-κB by inhibiting phosphorylation of the inhibitor of κB alpha (IκB-α), and distinctively inhibited the phosphorylation of ERK, JNK, and p38 MAPKs. In addition, the compounds increased the expression of antioxidant protein heme oxygenase (HO-1) by inducing nuclear translocation of nuclear factor erythroid 2-related factor 2 (Nrf2) through ERK and p38 MAPKs. Taken together, these results indicate that the synthetic glabridin derivatives exert strong anti-inflammatory effects in LPS-stimulated macrophages through MAPKs and NF-κB pathways, and support their development as potential therapeutics against inflammatory diseases.

## 1. Introduction

Inflammation is a critical process to protect the host from bacteria, viruses, and toxins, and plays a key role in removing the cause of such inflammation and restoring the damaged tissue. In general, when the cause is removed, the inflammation disappears [1]. However, if inflammation is maintained within the body for a long time due to environmental or physiological factors, the body continues to be in a state of chronic inflammation [1,2]. Chronic inflammation disrupts tissue homeostasis by inducing organelle dysfunction and cellular apoptosis within various tissues by the constant release of cytokines and/or chemokines [3,4]. Continuous exposure to these cytokines and/or chemokines is implicated in the pathogenesis of various inflammatory and inflammation-related diseases, such as autoimmune diseases including psoriasis, rheumatoid arthritis, and inflammatory bowel disease, metabolic diseases including non-alcoholic steatohepatitis (NASH) and type 2 diabetes, and even neurodegenerative disease [5,6]. Murine macrophage-like cell lines, such as RAW264.7 cells, are commonly used and are appropriate models for evaluating inflammatory responses under stimulation with lipopolysaccharide (LPS), which is the predominant outer component of gram-negative bacteria. The inflammatory responses are characterized by increased production of nitric oxide (NO), prostaglandin E_2_ (PGE_2_), tumor necrosis factor-α (TNF-α), and interleukin (IL)s [7,8]. Evaluating anti-inflammatory effects of potential therapeutics in such a system has been shown to be useful in the search for effective compounds against diverse inflammatory and inflammation-related diseases [9].

Glabridin is an isoflavan isolated from the roots extract of licorice (*Glycyrrhiza glabra*) [10]. Glabridin has been extensively studied as a natural compound with known anti-oxidative and anti-inflammatory activities, as well as with effects on the improvement of metabolic dysregulation [11]. However, glabridin has low stability and bioavailability, rendering it difficult to develop as a clinical therapeutic agent [12]. Previously, we performed a structure–activity relationship (SAR) study of glabridin and synthesized various glabridin derivatives with improved chemical stability and in vivo efficacy [13,14]. Among them, HSG4112 is currently at clinical phase 2 stage (NCT05197556) and HGR4113 is at clinical phase 1 stage (NCT05642377), and the (*S*)-enantiomer of HSG4112 (*S*)-HSG4112) is at a preclinical stage of development. Therefore, detailed characterization and understanding of the mechanism of the anti-inflammatory effects of these synthetic glabridin derivatives are needed.

Mitogen-activated protein kinase (MAPK) and nuclear factor kappa-light-chain-enhancer of activated B cells (NF-κB) signaling pathways both play pivotal roles in the mediation of the LPS-stimulated inflammatory response [15,16]. Three major MAPKs, which are extracellular signal-regulated kinase (ERK), c-Jun N-terminal kinase (JNK), and phosphorylated 38 (p38), are activated by phosphorylation and regulate inflammatory cytokine secretion in an LPS-stimulated RAW264.7 model [17,18,19,20,21]. All three MAPKs have been well studied to regulate inflammatory responses, reducing NO, IL-6, and TNF-α levels in LPS-induced models, respectively [22,23]. NF-κB is a transcription factor that is activated by the phosphorylation of nuclear factor kappa light polypeptide gene enhancer in B-cells inhibitor, alpha (IκB-α), and is localized to the nucleus and expresses various inflammatory response genes, including proinflammatory cytokines such as IL-1β, IL-6, and TNF-α, and proinflammatory factors such as inducible nitric oxygen synthase (iNOS) and cyclooxygenase-2 (COX-2) [15,24]. Suppressing the activation of MAPKs and NF-κB is thus an important step in mediating the anti-inflammatory effects of potential therapeutic compounds.

In contrast to pro-inflammatory responses, nuclear transcription factor E2-related factor 2 (Nrf2) is a major transcription factor that regulates anti-inflammatory and anti-oxidative response [25]. Nrf2 is localized to the nucleus under an oxidative stress condition and binds to the antioxidant response element (ARE) to induce gene expression of phase Ⅱ conjugation enzymes including heme oxygenase-1 (HO-1), a major antioxidant enzyme that regulates heme catabolism and cleaves heme to form biliverdin, carbon monoxide, and ferrous iron [25,26,27]. HO-1 and its product carbon monoxide can suppress the production of pro-inflammatory cytokines such as TNF-α, IL-1β and IL-6 [28,29]. The up-stream regulators of Nrf2 pathway include MAPKs (ERK, JNK, and p38) [30]. This pathway has been extensively researched for the search of therapeutic compounds as well.

In this study, we investigated the anti-inflammatory and anti-oxidative effects of the synthetic glabridin derivatives in LPS-induced RAW264.7 cells and examined their mechanisms on the major pro-inflammatory MAPK and NF-κB pathways and anti-inflammatory Nrf2 pathway.

## 2. Results

### 2.1. Effects of Compounds on LPS-Induced NO and PGE_2_ Production

We investigated whether glabridin and the synthetic glabridin derivatives—HSG4112, (*S*)-HSG4112, and HGR4113 (Figure 1)—exert overall anti-inflammatory effects in the LPS-stimulated RAW264.7 macrophage. RAW264.7 cells were pre-treated with glabridin and the synthetic glabridin derivatives at the indicated concentrations for 3 h before stimulation with LPS for 24 h, and the supernatant was collected. The indicated concentrations were used as the maximum concentration that did not visibly affect cell viability. Butein, a chalcone polyphenol first isolated from *Rhus verniciflua Stokes,* was used as a positive control, since it has various biological properties including anti-oxidative and anti-inflammatory effects, and has previously demonstrated such efficacy in RAW264.7 cell [31,32,33].

The level of NO and PGE_2_ was determined by the Griess reaction and ELISA, respectively. While LPS-stimulation significantly increased NO and PGE_2_ production, as shown in the control group; all compounds dose-dependently and significantly reduced such NO and PGE_2_ production (Figure 2A,B). All synthetic glabridin derivatives had superior inhibitory effects to glabridin and comparable inhibitory effects to butein at 20 µM concentration. As shown in Table 1, for the suppression of NO production, the IC_50_ values for glabridin, HSG4112, (*S*)-HSG4112, and HGR4113 were 9.36, 6.79, 3.85, and 11.32 μM, respectively. For the suppression of PGE_2_ production, the IC_50_ values for glabridin, HSG4112, (*S*)-HSG4112, and HGR4113 were 7.09, 3.55, 2.37, and 1.64 μM, respectively.

Next, the protein expressions of iNOS and COX-2, which produce NO and PGE_2_, respectively, were investigated. Cell lysate was harvested under the same experimental conditions as above, and the protein expression levels of iNOS and COX-2 were measured by Western blot. The expression levels of both iNOS and COX-2 were markedly increased by LPS-stimulation and were decreased in a dose-dependent manner by glabridin and the synthetic glabridin derivatives (Figure 2C–F).

### 2.2. Effects of Compounds on LPS-Induced Cytokines

The effects of glabridin and the synthetic glabridin derivates on the LPS-induced pro-inflammatory cytokines, including IL-1β, IL-6 and TNF-α, were examined. RAW264.7 cells were pre-treated with or without the indicated concentrations of compounds and then were stimulated with LPS. The mRNA expression of all three pro-inflammatory cytokines was markedly increased by LPS stimulation and was significantly decreased in a dose-dependent manner by glabridin and the synthetic glabridin derivatives: HSG4112, (*S*)-HSG4112, and HGR4113 (Figure 3A–C). All synthetic glabridin derivatives had superior inhibitory effects to glabridin and comparable inhibitory effects to butein at 20 µM concentration.

### 2.3. Effects of Compounds on NF-κB Signaling Pathway

To investigate the potential mechanism of the anti-inflammatory effects of the compounds, we examined the effects of glabridin and the synthetic glabridin derivates on NF-κB activation and DNA binding. NF-κB consists of two subunits (p50, p65) which are localized to the nucleus when activated. We performed Western blot to examine the translocation of NF-κB subunits into the nucleus. While both subunits p50 and p65 were localized into the nucleus by LPS-stimulation, glabridin and the synthetic glabridin derivatives —HSG4112, (*S*)-HSG4112, HGR4113—dose-dependently inhibited the nuclear translocation of NF-κB. Next, to identify which upstream pathway is affected in the NF-κB signaling, the activity of IκB-α was examined. Upon phosphorylation of IκB, IκB is degraded and separated from NF-κB, allowing NF-κB to translocate into the nucleus [24]. While IκB-α phosphorylation markedly increased with LPS-stimulation, glabridin and the synthetic glabridin derivatives decreased IκB-α phosphorylation in a dose-dependent manner (Figure 4A–D). In addition, nuclear DNA binding assay was performed to confirm the downstream effect of NF-κB. After NF-κB is translocated into the nucleus, it binds to DNA as a transcription factor. While nuclear NF-κB DNA binding markedly increased LPS-stimulation, glabridin and the synthetic glabridin derivatives significantly and dose-dependently decreased such binding (Figure 4E). These results indicate that glabridin and the synthetic glabridin derivatives inhibit the LPS-induced activation of NF-κB signaling by inhibiting phosphorylation of IκB-α.

### 2.4. Effects of Compounds on LPS-Induced MAPK Signaling

To further investigate the mechanism of the anti-inflammatory effects of the compounds, we examined the effects of glabridin and the synthetic glabridin derivates on MAPK signaling. We examined the activation of the three major MAPK (ERK, JNK, and p38) in RAW264.7 cells by quantifying phosphorylation through Western blot [34]. For the negative control, PD98059, SP600125, and SB203580 were used as inhibitors of ERK, JNK, and p38 MAPK, respectively. LPS stimulation markedly induced the phosphorylation of ERK, JNK, and p38 (Figure 5A–D). Under LPS-stimulation, the tested compounds showed a distinct effect on each of the MAPKS: glabridin suppressed phosphorylation of JNK and p38 (Figure 5A), HSG4112 suppressed phosphorylation of ERK (Figure 5B), (*S*)-HSG4112 suppressed phosphorylation of JNK and p38 (Figure 5C), and HGR4113 suppressed phosphorylation of JNK (Figure 5D). The total unphosphorylated forms of all MAPKs were unaffected by LPS and test compounds (Figure 5A–D). These results show that while glabridin and the synthetic glabridin derivatives inhibit at least one MAPK signaling under LPS-stimulation, the specific MAPK (ERK, JNK, and p38) involved is distinct for each compound.

### 2.5. Effects of Compounds on HO-1 Induction and Nrf2 Signaling

Intracellular inflammation can be caused by exogenous pathogens such as LPS but can also be caused by intracellular oxidative stress [35]. Nrf2 is a well-known anti-oxidative transcription factor that localizes into the nucleus to suppress cellular inflammatory conditions in an oxidative stress environment and induces transcription of antioxidant proteins such as HO-1 [36]. To investigate the potential anti-oxidative effects of the compounds, we examined whether glabridin and the synthetic glabridin derivatives induce HO-1 protein expression in RAW264.7 cells by Western blot. RAW264.7 cells were pretreated with or without the indicated concentrations of compounds or copper (CoPP) for 12 h. CoPP was used as a positive control to create an extreme oxidative stress environment and thus induce HO-1. We found that glabridin and the synthetic glabridin derivatives increase HO-1 protein expression in a dose-dependent manner in RAW264.7 cells (Figure 6A–D). Next, we investigated the mechanism of HO-1 induction by examining the effects of compounds on nuclear translocation of Nrf2 in RAW264.7 cells by Western blot. RAW264.7 cells were treated for 0.5, 1, and 1.5 h with each compound’s respective highest non-toxic concentrations. Butein was used as a positive control to induce nuclear translocation of Nrf2, which was previously reported in murine microglial cells [31]. Glabridin and the synthetic glabridin derivatives all increased nuclear Nrf2 expression and concomitantly decreased cytosolic Nrf2 expression (Figure 6E–H). These results demonstrate the anti-oxidative effects of glabridin and the synthetic glabridin derivatives through the Nrf2 signaling pathway.

### 2.6. Effects of Compounds on MAPK Signaling Involved in HO-1 Induction

To further investigate the mechanism of the anti-oxidative effects of the compounds, we examined the MAPKs upstream of Nrf2/HO-1 pathways, which are ERK, JNK, and p38 [30]. MAPK inhibitor assays were conducted by Western blot to determine which MAPK is involved in the expression of HO-1 in RAW264.7. Cells were pre-treated with respective MAPK inhibitor for 3 h, and then treated with each compound’s respective highest non-toxic concentration for 12 h. We found that for all compounds, p38 inhibitor (SB203580) and ERK inhibitor (PD98059) reduced the induction of HO-1, while the JNK inhibitor (SP600125) had no effect (Figure 7A–D). These results show that glabridin and the synthetic glabridin derivatives all induce the expression of HO-1 through p38 and ERK MAPKs.

## 3. Discussion

This study investigated the anti-inflammatory effects and mechanisms of the synthetic glabridin derivatives—HSG4112, (*S*)-HSG4112, and HGR4113—in comparison to glabridin in LPS-stimulated RAW264.7 murine macrophages. We found that glabridin and the synthetic glabridin derivates clearly and markedly suppress LPS-activated pro-inflammatory markers and cytokines, and inhibit LPS-induced activation of NF-κB and MAPK signaling pathways. In addition, all compounds induced the anti-inflammatory Nrf2 signaling pathway, increasing antioxidant HO-1 protein expression through distinct MAPKs.

Glabridin has an isoflavan structure and has various reported pharmacological activities, including anti-oxidative, anti-inflammatory, anti-atherogenic, energy-regulating, and neuroprotective effects [37]. In the previous study, we considered obesity as a chronic inflammatory condition and created biochemically stable synthetic glabridin derivatives (HSG4112, (*S*)-HSG4112, HGR4113) from the structure of glabridin and evaluated their efficacies through an in vivo SAR study [14]. Even though the backbone structure is similar, it is worthwhile to note that glabridin is an (*R*) enantiomer, while HSG4112 is a racemate and (*S*)-HSG4112 is an (*S*) enantiomer; for most cases of small molecular compounds, only one enantiomer is pharmacologically active [38]. In addition, HGR4113 has notable differences to glabridin, which are hydroxy-to-propoxy modification at the resorcinol ring at C-4 and the double bond hydrogenation at the pyranobenzene structure. Therefore, the anti-inflammatory effects and mechanisms of the synthetic glabridin derivatives could not be surmised and needed to be investigated.

The inflammatory response involves a number of key mediators—including NO, PGE_2_, IL-1β, IL-6, and TNF-α—which can also be used as clinical markers of diagnosis. While LPS induction dramatically increased the level of these markers as expected, the treatment of synthetic glabridin derivatives significantly reduced them in a degree greater than glabridin and comparable to Butein. This is indicative of how the reported functions of glabridin on these pro-inflammatory markers can be enhanced through chemical modifications [11,39].

LPS induction leads to the activation of pro-inflammatory pathways. In this study, we investigated two major signaling pathways—NF-κB and MAPK—to examine whether the tested compounds exhibit anti-inflammatory effects through them. NF-κB is a well-known protein that plays a pivotal role in the inflammatory response, and lies at the center of the pro-inflammatory cytokine response and NLR family pyrin domain containing 3 (NLRP3) inflammasome formation [18,40]. We found that the synthetic glabridin derivatives inhibit nuclear translocation of NF-κB by suppressing phosphorylation of IκB-α. However, it remains unknown whether the compounds directly engage IκB-α or any upstream signaling protein. Moreover, in MAPK phosphorylation, glabridin and (*S*)-HSG4112 inhibited JNK and p38, while HSG4112 inhibited ERK and HGR4113 inhibited JNK. This finding suggests that each compound has a notably distinctive mechanism of inhibiting the inflammatory response. There are reports of distinctive inhibitions of each of the MAPKs for the anti-inflammatory responses, while such inhibitions ultimately converge to the downstream effect of reducing the production of pro-inflammatory cytokines [41]. Whether each compound’s distinctive inhibition of MAPKs leads to differences in the inhibitory efficacy or mechanism of the inflammatory response is unknown, and the directly binding mechanistic target protein of each compound is also unknown [42,43,44]. The directly binding mechanistic target protein of each compound is yet unknown. Nevertheless, the synthetic glabridin derivatives showed overall superior anti-inflammatory effects compared to glabridin, which suggest that their efficacies would likely increase as therapeutic agents as well. There are pro-inflammatory mediates other than NF-κB. Cyclic AMP-responsive element-binding protein (CREB) and activator protein 1 (AP1) are also transcription factors that induce pro-inflammatory cytokines. These factors are activated by p38 and JNK MAPK pathway, respectively, by LPS-stimulation [45]. Signal transducer and activator of transcription 3 (STAT3) is another transcription factor activated by LPS-stimulation, and it can be activated by several cytokines mediating the expression of several acute-phase response genes [46,47]. Thus, further studies can investigate the compounds’ effects on CREB, AP-1, and STAT3-mediated induction of pro-inflammatory responses.

As opposed to pro-inflammatory signaling pathways, we evaluated the anti-inflammatory Nrf2 pathway to determine whether the synthetic glabridin derivatives exhibit anti-inflammatory and anti-oxidative effects. The Nrf2 pathway is a well-known pathway that induces the transcription of various anti-oxidative proteins including HO-1 [25,30,36,48]. Glabridin has been reported to regulate mitochondrial function and reduce ROS generation through the Nrf2/HO-1 signaling pathway [48,49]. The synthetic glabridin derivatives also induced nuclear translocation of Nrf2 and HO-1 expression. Of note, the MAPKs that are upstream of Nrf2 signaling were identified for each compound and were found to be ERK and p38 for all compounds, which suggests a common mechanism of action in mediating this pathway. However, further study is needed to investigate whether the synthetic glabridin derivatives induce the Nrf2/HO-1 pathway under oxidative stress condition by measuring ROS levels in cells or mitochondria.

In conclusion, our study demonstrated that the three prominent glabridin derivatives —HSG4112, (*S*)-HSG4112, and HGR4113—exhibit both anti-inflammatory and anti-oxidative effects in macrophages through canonical pathways involving MAPK, NF-κB, and Nrf2 signaling, as shown in Figure 8. These results provide support for their development as therapeutic agents against inflammatory and inflammation-related diseases.

## 4. Materials and Methods

### 4.1. Materials

Dulbecco’s modified Eagle’s medium (DMEM), fetal bovine serum (FBS), and various other tissue culture reagents were purchased from Thermo Fisher Scientific (Waltham, MA, USA). All other chemicals were obtained from Sigma-Aldrich Co. (St. Louis, MO, USA). Primary antibodies are anti-iNOS, sc-650; anti-COX-2, sc-1745; anti-IκB-α, sc-371; anti-p-IκB-α, sc-8404; anti-p50, sc-7178; anti-p65, sc-8008, from Santa Cruz Biotechnology (Dallas, TX, USA) anti-p-ERK, #9101; anti-ERK, #9102; anti-p-JNK, #9251; anti-JNK, #9252S; anti-p-p38, #9211; anti-p38, #9212S, from Cell Signaling Technology (Danvers, MA, USA) Secondary antibodies: anti-mouse, ap124p; anti-goat, ap106p; anti-rabbit, ap132p, Millipore. The enzyme-linked immunosorbent assay (ELISA) kit for PGE_2_ was purchased from R&D Systems (Minneapolis, MN, USA). The compounds glabridin, HSG4112, (*S*)-HSG4112, HGR4113 were provided by Glaceum Inc (Suwon, Republic of Korea).

### 4.2. Cell Culture and Viability Assay

RAW264.7 was purchased from American Type Culture Collection (ATCC, Manassas, VA, USA). RAW264.7 cells were cultured in 5 × 10^5^ cell/mL in DMEM medium, supplemented with 10% heat-inactivated FBS, penicillin G (100 units/mL), streptomycin (100 mg/mL), and L-glutamine (2 mM), and incubated at 37 °C in a humidified atmosphere containing 5% CO_2_. For all supernatant collections, the compounds were not washed out and the medium was not changed. Cell viability was measured with 3-[4,5-dimethylthiazol-2-yl]-2,5-diphenyltetrazolium bromide (MTT) assays, where 0.5 mg/mL of MTT was added to 200 μL of each cell suspension (1 × 10^5^ cell/mL in 96-well plates) for 4 h. The viability of compound-treated cells was measured through visual estimation to be similar to the control cells. Quantification was not performed.

### 4.3. NO Production

The nitrite concentration was used as an indicator of NO production and was measured with the Griess reaction. Supernatant (100 μL) was mixed with the Griess reagent at a 1:1 ratio (solution A, 222488; solution B, S438081; Sigma-Aldrich Co. (St. Louis, MO, USA)), and the absorbance at 525 nm was measured with ELISA plate reader.

### 4.4. PGE2 Assay

RAW264.7 cells were cultured in 24-well plates and incubated for 3 h with compounds before LPS (Sigma-Aldrich Co.) stimulation for 24 h. Supernatant (100 μL) PGE_2_ concentration was measured with an ELISA kit (R & D Systems).

### 4.5. Western Blot Analysis

RAW264.7 cells were harvested and centrifuged (16,000 rpm, 15 min). Cells were washed with PBS and lysed with 20 mM Tris-HCl buffer (pH 7.4) with protease inhibitor mixture (0.1 mM PMSF, 5 mg/mL pepstatin A, 5 mg/mL aprotinin, and 1 mg/mL chymostatin). Protein concentration was measured with Lowry Protein Assay Kit (P5626; Sigma-Aldrich Co.). Samples were placed on 12% sodium dodecyl sulfate-polyacrylamide gel electrophoresis (SDS-PAGE) and transferred to an enhanced chemiluminescence (ECL) nitrocellulose membrane (Bio-Rad, Hercules, CA, USA). The membrane was blocked with 5% skimmed milk and then incubated with the respective primary antibodies (Santa Cruz Biotechnology) and horseradish peroxidase-conjugated secondary antibodies, before ECL detection (Amersham Pharmacia Biotech, Amersham, UK).

### 4.6. Preparation of Cytosolic and Nuclear Fractions

RAW264.7 cells were homogenized in M-PER-Mammalian Protein Extraction Reagent (Thermo Fisher Scientific). The cytosolic fraction was acquired at 4 °C with centrifugation at 14,000× *g* for 5 min. Nuclear and cytoplasmic extracts were acquired with NE-PER Nuclear and Cytoplasmic Extraction Reagents (Pierce Biotechnology, Rockford, IL, USA). Cell lysis was performed at 4 °C by shaking for 15 min in RIPA Lysis and Extraction Buffer (Thermo Fisher Scientific). The final supernatant was collected by centrifugation at 16,000× *g* for 15 min.

### 4.7. DNA-Binding Activity of NF-κB

RAW264.7 cells were treated for 3 h with the compounds before LPS (1 μg/mL) stimulation for 30 min. The DNA-binding activity of NF-κB in the acquired nuclear extracts was measured with TransAM kit (Active Motif, Carlsbad, CA, USA).

### 4.8. Quantitative Real-Time Reverse Transcriptase PCR (qRT-PCR) Assay

Total RNA from RAW264.7 cells was isolated with Trizol (Invitrogen, Carlsbad, CA, USA) and quantified at 260 nm. Total RNA (1 μg) was reverse-transcribed with High-Capacity RNA-to-cDNA Kit (Applied Biosystems, Carlsbad, CA, USA) and cDNA was amplified with SYBR Premix Ex Taq Kit (TaKaRa Bio Inc., Shiga, Japan) and StepOnePlus Real-Time PCR (Applied Biosystems). qRT-PCR (20 μL) sample contained 10 μL SYBR Green PCR Master Mix, 0.8 μM primer, and the remaining diethyl pyrocarbonate (DEPC)-treated water. The primer sequences were designed with Primer Quest (Integrated DNA Technologies, Cambridge, MA, USA): IL-1β, forward 5′-A ATTGGTCATAGCCCGCACT-3′, reverse 5′-AAGCAATGTGCTGGTGCTTC-3′, IL-6, forward 5′-ACTTCACAAGTCGGAGGCTT-3′, reverse 5′- TGCAAGTGCATCATCGTTGT-3′, TNF-α, forward 5′-CCAGACCCTCACACTCACAA-3′, reverse 5′-A CAAGGTACAACCCATCGGC-3′, GAPDH, forward 5′-ACTTTGGTATCGTGGAAGGACT-3′, reverse 5′- GTAGAGGCAGGGATGATGTTCT-3′. Data were analyzed with Thermal Cycler and StepOne software (Applied Biosystems) and the comparative CT method was used to measure the relative gene expression using GAPDH as the endogenous control.

### 4.9. Statistical Analysis

Data are expressed as the mean ± SD. Three independent experiments were performed per assay. One-way analysis of variance (ANOVA) with Dunnett’s multiple comparison tests was used to compare the groups. Statistical analysis was performed with GraphPad Prism software, version 9.40 (GraphPad Software Inc., San Diego, CA, USA).

## Figures and Tables

**Figure 1 molecules-28-02135-f001:**
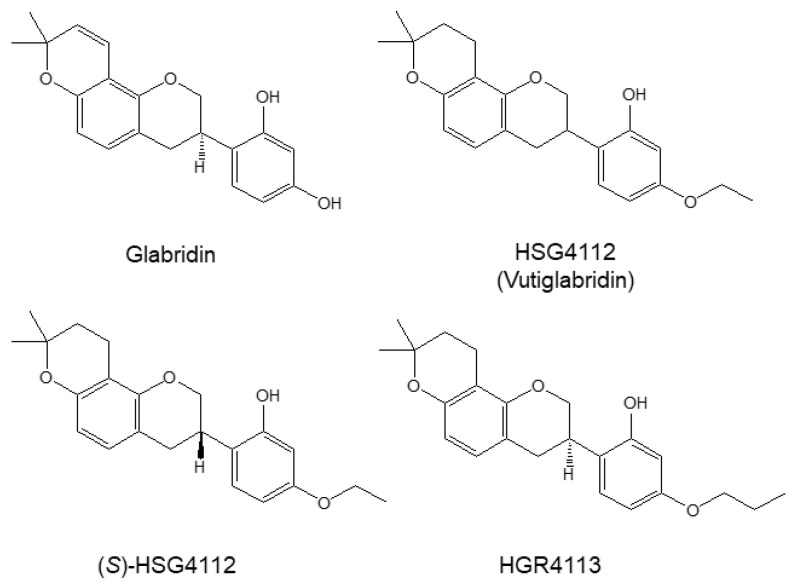
Structures of glabridin and the synthetic glabridin derivatives: HSG4112, (*S*)-HSG4112, HGR4113. Chemical structures of glabridin and the synthetic glabridin derivatives: Glabridin, HSG4112, (*S*)-HSG4112, HGR4113.

**Figure 2 molecules-28-02135-f002:**
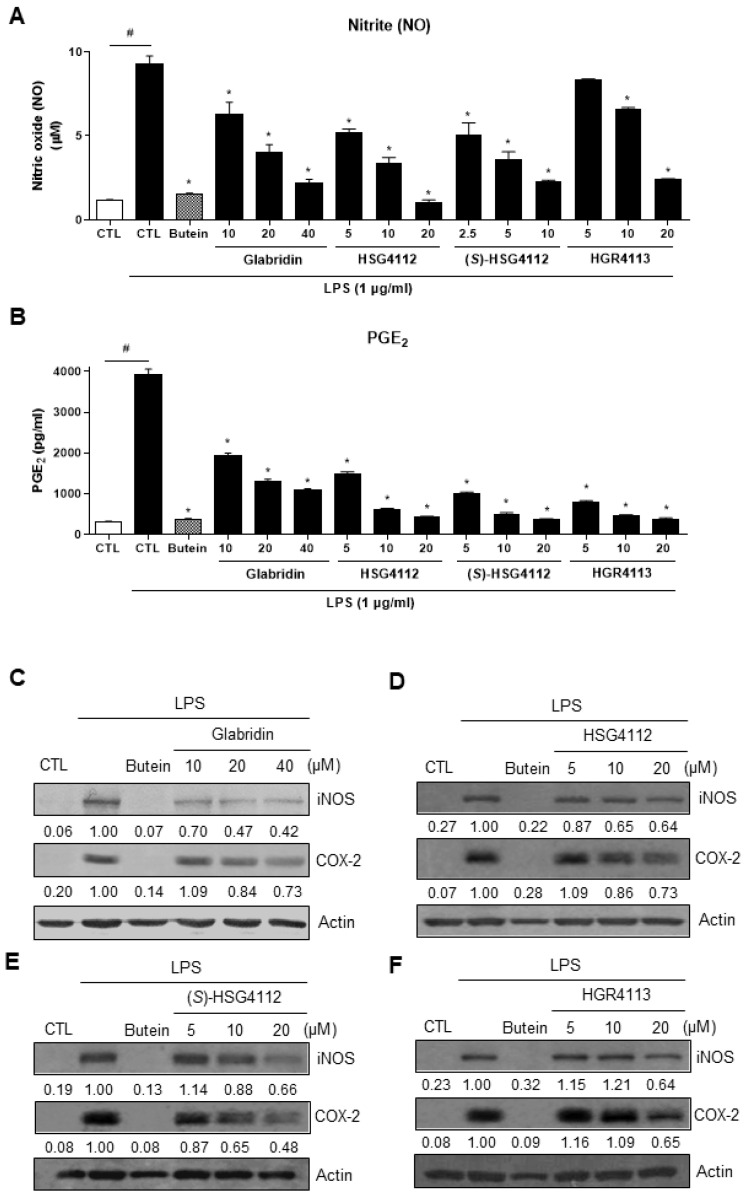
Effect of compounds on LPS-induced NO, PGE_2_, iNOS, and COX-2 in RAW264.7 cells. RAW264.7 cells were pre-treated with or without the indicated concentrations of compounds for 3 h and then stimulated with LPS (1 μg/mL) for 24 h. Butein (10 μM) was used as positive control with known inhibitory effect against LPS-stimulation (hatched pattern column). Control (CTL) groups were not treated with any compounds nor LPS (white column). (**A**) NO concentration was determined using Griess reaction. (**B**) PGE_2_ was measured by ELISA. Values shown are means ± SD of three independent experiments. ^#^
*p* < 0.05 in LPS-treated CTL vs. CTL, and * *p* < 0.05 vs. LPS-treated CTL group. (**C**–**F**) The protein levels of iNOS and COX-2 were determined by Western blot. The experiment was repeated three times. The protein expressions in the Western blot were quantitatively presented at the bottom of each band as relative values to the actin expression.

**Figure 3 molecules-28-02135-f003:**
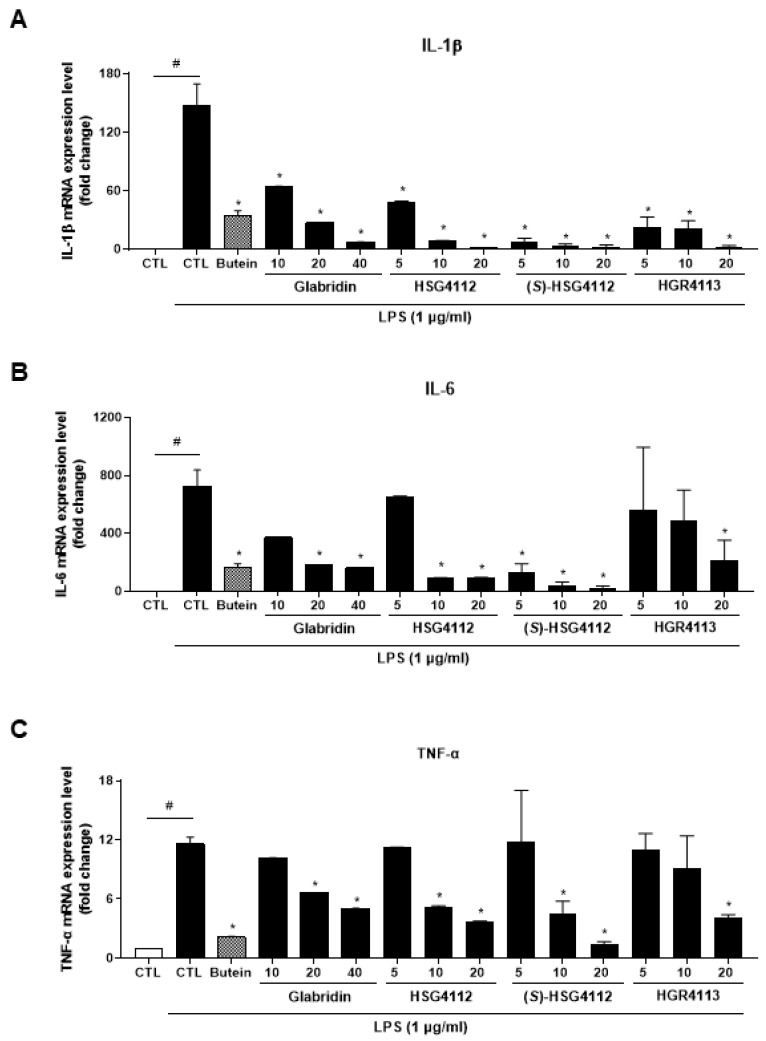
Effects of compounds on LPS-induced IL-1β, IL-6, TNF-α expression in RAW264.7 cells. RAW264.7 cells were pre-treated with or without the indicated concentrations of compounds for 3 h and then stimulated with LPS (1 μg/mL) for 6 h. Butein (10 μM) was used as positive control with known inhibitory effect against LPS-stimulation (hatched pattern column). CTL groups were not treated with any compounds nor LPS (white column). The mRNA levels of (**A**) IL-1β, (**B**) IL-6, and (**C**) TNF-α were determined by qPCR. Values shown are means ± SD of three independent experiments. ^#^
*p* < 0.05 in LPS-treated CTL vs. CTL, and * *p* < 0.05 vs. LPS-treated CTL group.

**Figure 4 molecules-28-02135-f004:**
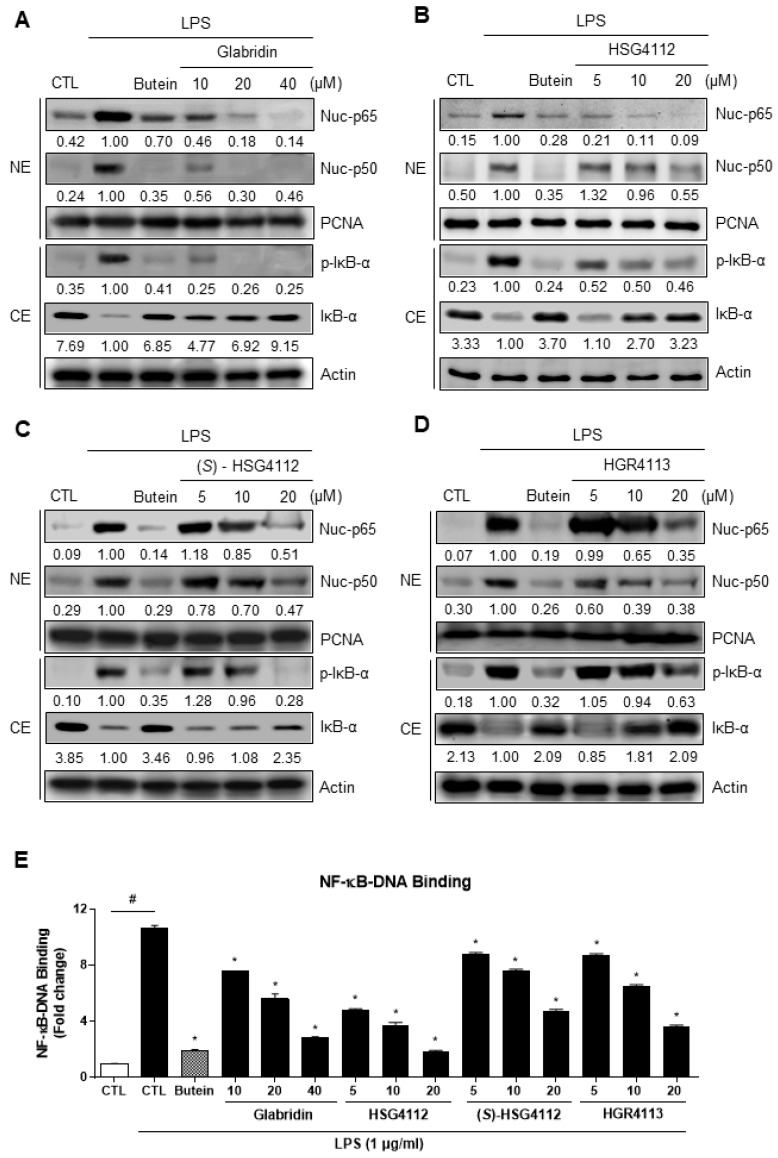
Effects of compounds on LPS-induced NF-κB nuclear translocation and DNA binding in the nucleus in RAW264.7 cells. RAW264.7 cells were pre-treated with or without the indicated concentrations of compounds for 3 h and then stimulated with LPS (1 μg/mL) for 1 h. Butein (10 μM) was used as positive control with known inhibitory effect against LPS-stimulation (hatched column). CTL groups were not treated with any compounds nor LPS (white column). (**A**–**D**) Cytosolic extracts (CE) were isolated and the levels of p-IκB-α and IκB-α in each fraction were determined by Western blot. Nuclear extracts (NE) were isolated and the levels of p65 and p50 in each fraction were determined by Western blot. (**E**) NF-κB ELISA kit (Active Motif) was used on the nuclear extracts to determine the degree of NF-κB DNA binding. PCNA was used as nuclear lysate control and actin was used as cytosolic lysate control. The experiment was repeated three times. The protein expressions in the Western blot were quantitatively presented at the bottom of each band as relative values to the PCNA and actin expression, respectively. Values shown are means ± SD of three independent experiments ^#^
*p* < 0.05 in LPS-treated CTL vs. CTL, and * *p* < 0.05 vs. LPS-treated CTL group. NE = nuclear extracts, CE = cytosolic extracts.

**Figure 5 molecules-28-02135-f005:**
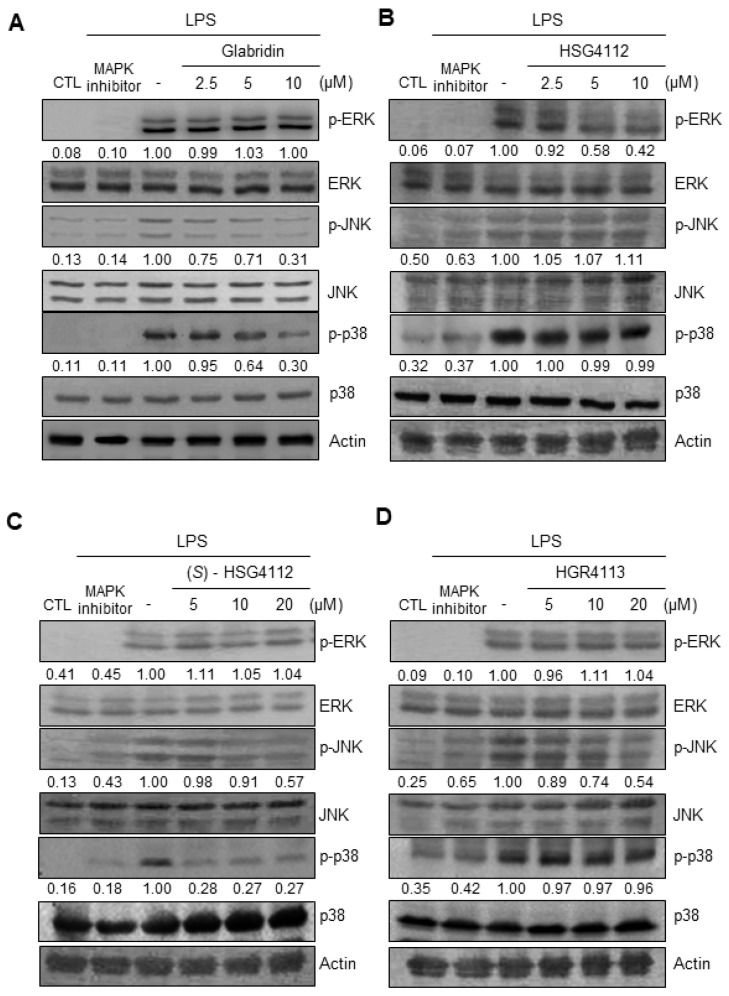
Effects of compounds on LPS-induced phosphorylation of ERK, JNK, and p38 MAPK in RAW264.7 cells. Lysates were prepared from RAW264.7 cells pre-treated with or without the indicated concentrations of compounds for 3 h and then stimulated with LPS (1 μg/mL) for 30 min. (**A**–**D**) The protein levels of the phosphorylated form and non-phosphorylated form were determined by Western blot. CTL groups were not treated with any compounds. PD98059, SP600125, SB203580 were used as positive controls to inhibit the phosphorylation of ERK, JNK, and p38, respectively. The experiment was repeated three times. The protein expressions in the Western blot were quantitatively presented at the bottom of each band as relative values to the actin expression.

**Figure 6 molecules-28-02135-f006:**
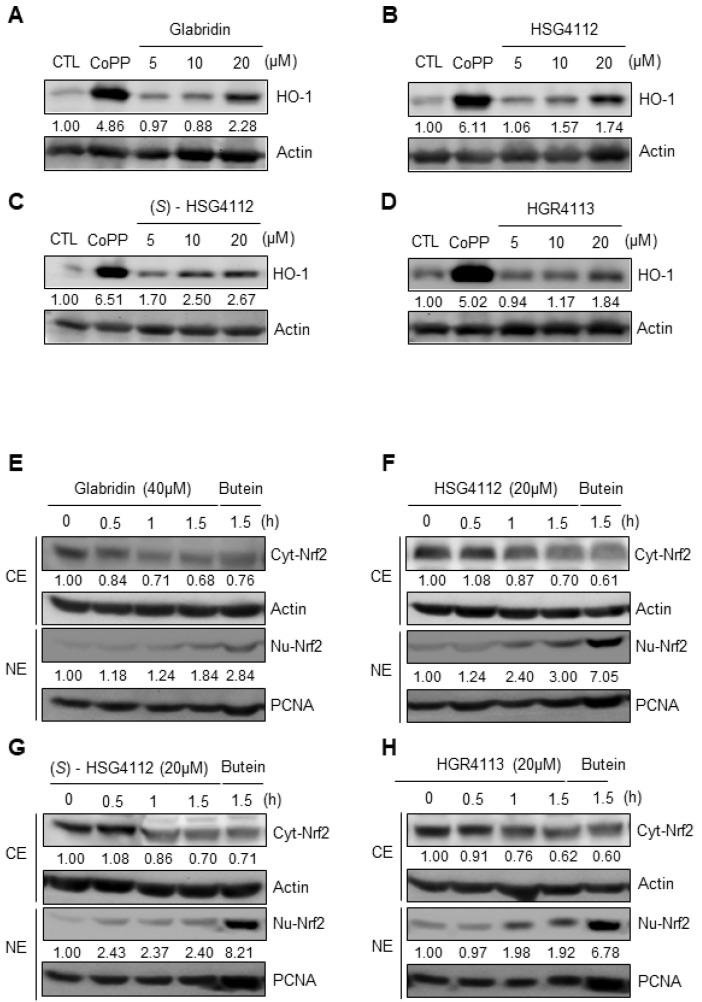
Effects of compounds on HO-1 expression and Nrf2 nuclear translocation in RAW264.7 cells. (**A**–**D**) RAW264.7 cells were treated with or without the indicated concentrations of compounds for 12 h. Copper (CoPP) was used as positive control. The protein level of HO-1 was determined by Western blot. CTL groups were not treated with any compounds. The protein expressions in the Western blot were quantitatively presented at the bottom of each band as relative values to the actin expression. (**E**–**H**) RAW264.7 cells were treated with the highest non-toxic concentrations of compounds for 0, 0.5, 1, and 1.5 h. Nuclear and cytosolic extracts were isolated and the levels of Nrf2 in each fraction were determined by Western blot. Butein (10 μM) was used as positive control. PCNA was used as nuclear lysate control and actin was used as cytosolic lysate control. The experiment was repeated three times. The protein expressions in the Western blot were quantitatively presented at the bottom of each band as relative values to the PCNA and actin expression, respectively. NE = nuclear extracts, CE = cytosolic extracts.

**Figure 7 molecules-28-02135-f007:**
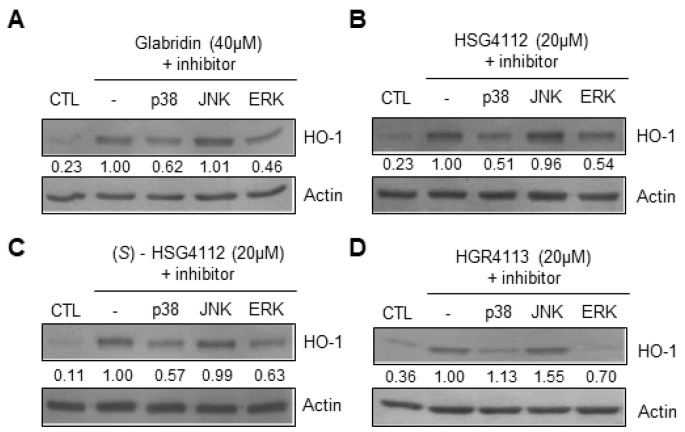
Effects of compounds on MAPKs involved in HO-1 expression in RAW264.7 cells. RAW264.7 cells were pre-treated with or without MAPK inhibitors for p38 (SB203580), JNK (SP600125), or ERK (PD98059) for 3 h, and then treated with the highest non-toxic concentration of each compound for 12 h. (**A**–**D**) The protein level of HO-1 was determined by Western blot. CTL groups were not treated with any compounds. The protein expressions in the Western blot were quantitatively presented at the bottom of each band as relative values to the actin expression.

**Figure 8 molecules-28-02135-f008:**
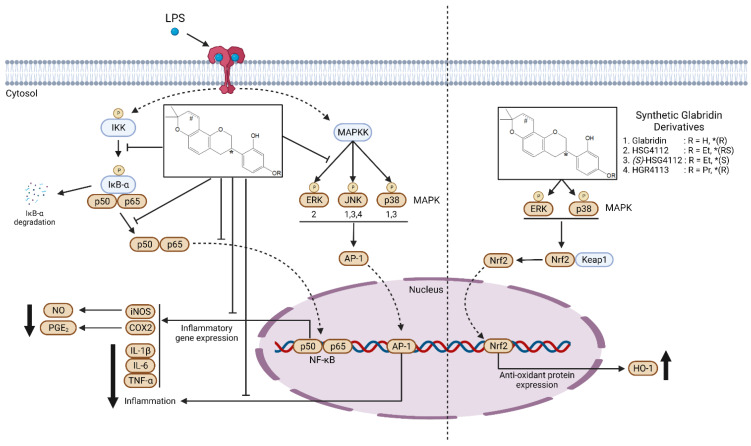
Molecular mechanism model for the anti-inflammatory effect of synthetic glabridin derivatives in RAW264.7 cells. Synthetic glabridin derivatives have anti-inflammatory effects through NF-kB and MAPKs signaling pathways under LPS stimulation. Compounds also induce the antioxidant protein HO-1 through Nrf2 pathway. In the chemical structure of the synthetic glabridin derivatives, * indicates the stereocenter for the optical isomers, R indicates the functional group, and ^#^indicates a double bond, only in glabridin. The number written under the MAPK (left panel) indicates which compound among the synthetic glabridin derivatives acts distinctively through the three MAPKs.

**Table 1 molecules-28-02135-t001:** Summary of the effects of the compounds in LPS-stimulated RAW264.7 cells.

Compound	NO (IC_50,_ μM)	PGE_2_ (IC_50,_ μM)	Cytokine mRNA Expression Inhibition	NF-κB Pathway Inhibition	MAPK Pathway Inhibition	HO-1 Induction by MAPK
**Glabridin**	9.36	7.09	IL-1β, IL-6, TNF-α	p-IκB-α	JNK, p38	p38, ERK
**HSG4112**	6.79	3.55			ERK	
**(*S*)-HSG4112**	3.85	2.37			JNK, p38	
**HGR4113**	11.32	1.64			JNK	

## Data Availability

The data presented in this study are available on request from the corresponding author.

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
