# Peer review of "Synthetic Glabridin Derivatives Inhibit LPS-Induced Inflammation via MAPKs and NF-κB Pathways in RAW264.7 Macrophages"

_molecules, 2023, doi:10.3390/molecules28052135_

Round 1
Reviewer 1 Report
Authors investigated the anti-inflammatory 22 effects of the glabridin derivatives in lipopolysaccharide (LPS)-stimulated RAW264.7 macrophages. This study is well designed and represented.
Author Response
Thank you for your time and your kind review.
Reviewer 2 Report
The authors investigated the anti-inflammatory effects of the glabridin derivatives in lipopolysaccharide (LPS)-stimulated RAW264.7 macrophages. They found that the synthetic glabridin derivatives significantly and dose-dependently suppressed the production of nitric oxide (NO) and prostaglandin E2 (PGE2) and decreased the level of inducible nitric oxygen synthase (iNOS) and cyclooxygenase-2 (COX-2) and the expression of pro-inflammatory cytokines interleukin-1β (IL-1β), IL-6, and tumor necrosis factor alpha (TNF-α). The synthetic glabridin derivatives inhibited the nuclear translocation of the NF-κB by inhibiting phosphorylation of inhibitor of κB alpha (IκB-α), and distinctively inhibited the phosphorylation of ERK, JNK, and p38 MAPKs. In addition, the compounds increased the expression of antioxidant protein heme oxygenase (HO-1) by inducing nuclear translocation of nuclear factor erythroid 2-related factor 2 (Nrf2) through ERK and p38 MAPKs. The manuscript is well written and sufficiently descriptive and I recommend to accept the manuscript after minor revision.
1. Please quantify the the western blot against LPS-induced control and recalculate the fold changes.
2. Please use separate marker for nuclear and cytoplasmic extract like Histone and tubulin to comment own the translocation restriction.
Author Response
Thank you for your comment, revision details are put in word file.

Reviewer 3 Report
In their work, the authors presented interesting results showing the anti-inflammatory and cytoprotective effects of glabridin derivatives in macrophages. They demonstrated that the synthetic glabridin derivatives suppressed NO production, and decreased the levels of iNOS, COX-2, and pro-inflammatory cytokines. Besides that, glabridin derivatives inhibited the ERK, JNK, and p38 MAPKs phosphorylation and activated antioxidant proteins. Overall, studies confirmed the therapeutic potential of novel compounds and their potential implication in inflammatory-mediated disease.
The manuscript is well-written and the research conducted is well-explained. The authors performed analyses that are valuable as a sort of first-phase pharmacological screening of the mechanism of action for potential candidates for new therapeutics. Results are well presented and described. Even though the methodology is very simple, the research has been performed and presented in a justified way. Obtained data provide a good basis for developing a more advanced methodology to confirm the mechanism of action of the compounds and to consider further their potential metabolites and potential mechanism of action.
I am pleased to recommend the work for publication in the Molecules.
Author Response
Thank you very much for your concise summary of our work and your time in reviewing our manuscript. We appreciate your kind comments. Thank you.
Reviewer 4 Report
The authors have shown that Glabridin and its derivatives inhibit the inflammatory response elicited by LPS. These compounds also inhibit the activation of the MAPK and NF-kB pathways.
There are several points that need to be addressed.
1) There are two notations for pretreatment time: 3 hours and 30 minutes.
Which one is correct?
Was LPS added to the medium with chemicals and collected 24 hours later or were the chemicals washed out before adding LPS?
2) Why is the concentration of (S)-HSG4112 different here only in A of Figure 2 from the others? In other figures, it is 5, 10, and 20, but only here it is 2.5, 5, and 10.
(3) In Figure 2, there is no description of what the colors on the graph mean. It should be mentioned.
(4) The chemicals under consideration are shown to inhibit the LPS response and MAPK.
Authors claimed that they showed chemicals inhibited LPS response via MAPK.
This reviewer is concerned because the effect of each drug on MAPKs does not necessarily correlate with its effect on the inflammatory response.
Do we need to show that each MAPK inhibitors have a similar or different effect on inflammatory response induced by LPS? Or, if it has already been found in a previous report as described in introduction section, that point should be clearly stated.
5) Why change from CoPP (A-D) to Butein (E-H) as positive control in Figure 6?
One would expect the same results, but would the effects of both on HO-1 and Nrf2 have been different?
6) I think it is necessary to mention, for example, the name of the p38 inhibitor (in the figure, p38) in the description of Fig. 7.
Author Response

(The authors gave the same response as above.)
